# Presenting Features and Early Mortality from SARS-CoV-2 Infection in Cancer Patients during the Initial Stage of the COVID-19 Pandemic in Europe

**DOI:** 10.3390/cancers12071841

**Published:** 2020-07-08

**Authors:** David J. Pinato, Alvin J. X. Lee, Federica Biello, Elia Seguí, Juan Aguilar-Company, Anna Carbó, Riccardo Bruna, Mark Bower, Gianpiero Rizzo, Sarah Benafif, Carme Carmona, Neha Chopra, Claudia Andrea Cruz, Francesca D’Avanzo, Joanne S. Evans, Myria Galazi, Isabel Garcia-Fructuoso, Alessia Dalla Pria, Thomas Newsom-Davis, Diego Ottaviani, Andrea Patriarca, Roxana Reyes, Rachel Sharkey, Christopher C. T. Sng, Yien Ning Sophia Wong, Daniela Ferrante, Lorenza Scotti, Gian Carlo Avanzi, Mattia Bellan, Luigi Mario Castello, Javier Marco-Hernández, Meritxell Mollà, Mario Pirisi, Isabel Ruiz-Camps, Pier Paolo Sainaghi, Gianluca Gaidano, Joan Brunet, Josep Tabernero, Aleix Prat, Alessandra Gennari

**Affiliations:** 1Department of Surgery & Cancer, Imperial College London, Hammersmith Hospital, London W12 0HS, UK; joanne.evans@imperial.ac.uk; 2Department of Oncology, University College London Hospitals, London NW1 2PG, UK; a.j.lee@ucl.ac.uk (A.J.X.L.); sarah.benafif@nhs.net (S.B.); neha.chopra@nhs.net (N.C.); myria.galazi@nhs.net (M.G.); d.ottaviani@ucl.ac.uk (D.O.); christopher.sng@nhs.net (C.C.T.S.); Sophia.wong@nhs.net (Y.N.S.W.); 3Department of Translational Medicine, Division of Oncology, University of Piemonte Orientale and Maggiore della Carita’ Hospital, 28100 Novara, Italy; federica.biello@maggioreosp.novara.it (F.B.); francesca.davanzo@maggioreosp.novara.it (F.D.); alessandra.gennari@med.uniupo.it (A.G.); 4Department of Medical Oncology, Hospital Clinic, 08036 Barcelona, Spain; SEGUI@clinic.cat (E.S.); CACRUZ@clinic.cat (C.A.C.); RMREYES@clinic.cat (R.R.); alprat@clinic.cat (A.P.); 5Department of Medical Oncology, Vall d’Hebron University Hospital and Institute of Oncology (VHIO), 08035 Barcelona, Spain; jaguilar@vhio.net (J.A.-C.); jtabernero@vhio.net (J.T.); 6Department of Infectious Diseases, Vall d’Hebron University Hospital, 08035 Barcelona, Spain; iruiz@vhebron.net; 7Department of Medical Oncology, Catalan Institute of Oncology, University Hospital Josep Trueta, 17007 Girona, Spain; acarbo@iconcologia.net (A.C.); ccarmona@iconcologia.net (C.C.); igarciaf.girona.ics@gencat.cat (I.G.-F.); jbrunet@iconcologia.net (J.B.); 8Department of Translational Medicine, Division of Haematology, University of Piemonte Orientale and Maggiore della Carita’ Hospital, 28100 Novara, Italy; riccardo.bruna@maggioreosp.novara.it (R.B.); andrea.patriarca@maggioreosp.novara.it (A.P.); rachel.sharkey@chelwest.nhs.uk (R.S.); gianluca.gaidano@med.uniupo.it (G.G.); 9Department of Oncology and National Centre for HIV Malignancy, Chelsea & Westminster Hospital, London SW109NH, UK; m.bower@imperial.ac.uk (M.B.); a.dalla-pria17@imperial.ac.uk (A.D.P.); tom.newsom-davis@chelwest.nhs.uk (T.N.-D.); 10Department of Medical Oncology Unit, Fondazione IRCCS Policlinico San Matteo, 27100 Pavia, Italy; g.rizzo@smatteo.pv.it; 11Department of Translational Medicine, Unit of Cancer Epidemiology, CPO-Piemonte, University of Piemonte Orientale, 28100 Novara, Italy; daniela.ferrante@uniupo.it; 12Department of Translational Medicine, University of Piemonte Orientale and Maggiore della Carita’ Hospital, 28100 Novara, Italy; lorenza.scotti@uniupo.it; 13Department of Translational Medicine, Divisions of Internal and Emergency Medicine, University of Piemonte Orientale and Maggiore della Carita’ Hospital, 28100 Novara, Italy; giancarlo.avanzi@uniupo.it (G.C.A.); mattia.bellan@med.uniupo.it (M.B.); luigi.castello@med.uniupo.it (L.M.C.); mario.pirisi@med.uniupo.it (M.P.); pierpaolo.sainaghi@med.uniupo.it (P.P.S.); 14Department of Internal Medicine, Hospital Clinic, 08035 Barcelona, Spain; JMARCO@clinic.cat; 15Department of Radiation Oncology, Hospital Clinic, 08035 Barcelona, Spain; MOLLA@clinic.cat; 16Translational Genomics and Targeted Therapies in Solid Tumors, IDIBAPS, 08035 Barcelona, Spain

**Keywords:** COVID-19, coronavirus, SARS-CoV-2, cancer, survival, outcomes, mortality

## Abstract

We describe the outcomes in cancer patients during the initial outbreak of the COVID-19 in Europe from the retrospective, multi-center observational OnCovid study. We identified 204 cancer patients from eight centers in the United Kingdom, Italy, and Spain aged >18 (mean = 69) and diagnosed with COVID-19 between 26 February and 1 April 2020. A total of 127 (62%) were male, 184 (91%) had a diagnosis of solid malignancy, and 103 (51%) had non-metastatic disease. A total of 161 (79%) had >1 co-morbidity. A total of 141 (69%) patients had >1 COVID-19 complication. A total of 36 (19%) were escalated to high-dependency or intensive care. A total of 59 (29%) died, 53 (26%) were discharged, and 92 (45%) were in-hospital survivors. Mortality was higher in patients aged >65 (36% vs. 16%), in those with >2 co-morbidities (40% vs. 18%) and developing >1 complication from COVID-19 (38% vs. 4%, *p* = 0.004). Multi-variable analyses confirmed age > 65 and >2 co-morbidities to predict for patient mortality independent of tumor stage, active malignancy, or anticancer therapy. During the early outbreak of SARS-CoV-2 infection in Europe co-morbid burden and advancing age predicted for adverse disease course in cancer patients. The ongoing OnCovid study will allow us to compare risks and outcomes in cancer patients between the initial and later stages of the COVID-19 pandemic.

## 1. Introduction

Coronavirus disease 2019 (COVID-19), the viral infection caused by SARS-CoV-2, recognizes a spectrum of clinical entities ranging from a mild flu-like illness to life-threatening forms with acute respiratory compromise [1]. Advancing age and presence of comorbidities appear to increase the risk of severe COVID-19 [2]. During the early stages of the COVID-19 outbreak in the Hubei province in China, cancer patients were noted to have a 6.2-fold difference in mortality compared to healthy individuals (5.6% vs. 0.9%) [3]. Mortality among cancer patients who develop COVID-19 has subsequently been reported to range between 9% and 33%, however many of these studies were either non-European studies or focused on the later stages of the COVID-19 pandemic in Europe [4,5,6,7,8,9]. The relative contribution of cancer over other co-morbidities and clinical factors in influencing outcome from SARS-CoV-2 infection has not been fully elucidated.

COVID-19 at present is an ongoing global pandemic with close to 10 million confirmed cases and 500,000 confirmed deaths from COVID-19 in more than 200 countries as of June 2020 [10]. Non-pharmaceutical interventions including lockdowns have helped to reduce transmission of this disease [11]. The ongoing wellbeing of cancer patients and safe delivery of anticancer treatment remains a pressing issue with the continuing COVID-19 pandemic and the threat of a “second wave”.

With the declaration of SARS-CoV-2 as a pandemic pathogen, urgent measures were implemented on a global scale to protect cancer patients from morbidity and mortality including limiting hospital attendance and deferring systemic anticancer treatment [12]. Whilst dictated by the necessary adoption of a precautionary principle in the context of rapidly escalating viral transmission, these measures rested on the assumption of a detrimental effect of cancer and active anticancer therapy on outcomes from SARS-CoV-2 infection [6].

To address the gaps in knowledge on whether the presence of cancer preconditions patients towards a more adverse clinical course from SARS-CoV-2 infection [13], we designed OnCovid, a retrospective, observational multi-center study aimed at describing the natural history and outcomes from SARS-CoV-2 infection in European cancer patients. This is a point of greater consequence given the need for ongoing risk-stratification of cancer patients and the ongoing need to deliver anticancer treatment during the COVID-19 pandemic including any “second waves”.

Here, we report an analysis of the study repository, focusing on the early phase of the COVID-19 pandemic in Europe. The early phase of the pandemic in Europe was characterized by steps taken to protect cancer patients including shielding or isolating cancer patients and a reduction in delivery of anticancer treatments due to their presumed deleterious effects [14,15]. Moreover, healthcare resources are finite and were reallocated towards dealing with the immediate effects of COVID-19 rather than ongoing delivery of anticancer treatment during the initial outbreak of COVID-19 in Europe [16]. Limited intensive care unit capacity may also have restricted escalation of care of cancer patients who contracted COVID-19. Consequently, there was a difference in provision of care and exposure risk to COVID-19 in cancer patients between the early and later phase of the COVID-19 pandemic. Further outputs from this OnCovid study will allow us to compare differences in outcomes in cancer patients between the early and later stages of the COVID-19 pandemic in Europe given the changes during the later stage of the pandemic in Europe including increased delivery of anticancer treatments and change in social distancing or shielding policies.

## 2. Results

### 2.1. Demographics and Oncological Features

Between February 26th to April 1st 2020, we identified 204 patients with confirmed SARS-CoV-2 infection and cancer at the eight centers surveyed in the United Kingdom (*n* = 97, 47.5%), Italy (*n* = 56, 27.4%), and Spain (*n* = 51, 25%). This number included all the consecutive referrals received by acute oncology and/or emergency/internal medicine services during the accrual period. Demographic and clinical features of the patient population are described in Table 1. The majority of patients were men (62.3%) with a mean (±SD) age of 69.3 (±13) years (range 21–99). Most patients (*n* = 184, 90.2%) carried a diagnosis of solid malignancy with advanced stage occurring in 82 patients (44.3%); genito-urinary cancers represented the commonest primary site (*n* = 43, 21.1%). Most patients had received prior anticancer treatment, in the adjuvant/neoadjuvant setting in solid tumors (*n* = 49, 24%), in the curative setting in hematological malignancies (*n* = 13, 68.4%) or with palliative intent (*n* = 61, 29.9%), whereas 48 patients (23.5%) were treatment naïve. At COVID-19 diagnosis, 128 patients (62.8%) had evidence of active malignancy and 102 (50.0%) were on active anticancer therapy, mostly with palliative intent (*n* = 69, 67.7%). Co-morbid conditions were documented in 161 patients (78.9%), the most prevalent being hypertension (43.1%), diabetes mellitus (22.6%), and cardiovascular diseases (21.5%). In total, 107 patients (52.5%) had >1 co-morbidity. At COVID-19 diagnosis, 27 patients (13.2%) were assuming corticosteroids at the dose of >10 mg of prednisone equivalent.

### 2.2. Features of COVID-19 Disease

The most common presenting symptoms of SARS-CoV-2 infection were fever (*n* = 136, 66.7%), cough (*n* = 119, 58.3%), and dyspnea (*n* = 81, 39.7%). Mean body temperature at presentation was 37.9 °C (SD ± 0.9). SARS-CoV-2 was community-acquired in 153 patients (75%) and complicated a pre-existing hospital admission in another 51 (25%). Mean time from onset of symptoms to presentation was 3.8 days (SD ± 4.5). Radiologic investigations including a chest X-ray (CXR) and/or computerized tomography (CT) were performed at the discretion of the treating physician in 187 (91.7%) patients. Acute abnormalities were found in 103 out of 147 patients with a baseline CXR available (72.0%) and in 49 out of 50 patients with a baseline CT (98%). Bilateral ground-glass/reticulo-nodular changes were the most commonly observed pattern on CXR (*n* = 67, 46.6%) and on CT (*n* = 47, 94%).

In a subset of patients, we were able to compare laboratory findings at COVID-19 diagnosis with those obtained during the most recent oncological follow-up preceding SARS-Cov-2 infection. As shown in Table 2, comparison of routine blood tests across the two timepoints demonstrates an acute, significant shift in the circulating levels of albumin, C-reactive protein, alanine amino-transferase (ALT), sodium, hemoglobin, lymphocyte, and platelet counts at COVID-19 diagnosis secondary to the acute illness (Table 2).

### 2.3. Outcomes

Throughout the observation period, the majority of patients (*n* = 141, 69%) developed at least one complication from COVID-19, the most common being acute respiratory failure (*n* = 78, 38.2%) followed by ARDS (*n* = 49, 24.0%). Fifty-one out of 204 patients (25%) had evidence of an uncomplicated illness. The majority of patients were treated in the context of ward-based care (*n* = 186, 91.2%), 36 of whom (19.3%) required escalation to intensive/sub-intensive care. In 15 cases (7.4%) admission to hospital was not deemed necessary and patients were managed with domiciliary self-isolation. The median length of hospitalization in admitted patients was 8 days (IQR 5–13) and median permanence in intensive-sub/intensive care unit was 5 days (IQR 3–10). Oxygen therapy was administered to 128 patients (62.8%) including high-flow delivery in 57 (44.5%). Mechanical ventilation was initiated in 18 patients (8.8%) including non-invasive ventilation (*n* = 10, 55.6%) and endotracheal intubation (*n* = 8, 44.4%). None of the patients received extracorporeal membrane oxygenation (ECMO). Empirical therapy for COVID-19 was initiated in 138 patients (67.7%) and included antibiotics (*n* = 107, 77.5%), chloroquine/hydroxychloroquine (*n* = 84, 60.9%) and anti-virals (*n* = 63, 30.9%) such as lopinavir/ritonavir (*n* = 41, 29.7%), remdesivir (*n* = 9, 4.4%), and darunavir/cobicistat (*n* = 6, 3%). At time of censoring on April 6th 2020, of the 204 patients accrued, 59 had died (28.9%), 92 (45.1%) were in-hospital survivors, and 53 (26%) were discharged from hospital. The mortality rate stratified by country was 25% for Italian (*n* = 14/56), 29.4% for Spanish (*n* = 15/70), and 30.9% for UK centers (*n* = 30/97).

### 2.4. Factors Associated with Mortality from COVID-19 in Cancer Patients

We observed a significantly higher mortality rate in patients aged ≥65 years (79.7% vs. 20.3%, *p* = 0.003; HR 2.6, 95%CI 1.4–5.0) and in those with ≥2 pre-existing comorbidities (71.2% vs. 28.8%, *p* = 0.003; HR 2.4, 95%CI 1.3–4.2, Figure 1, Table 3). We also noted differential distribution in a number of laboratory parameters collected at COVID-19 diagnosis in relationship with patients’ mortality (Table 3). Multivariable analysis confirmed age ≥ 65 years and presence of ≥2 pre-existing co-morbidities to be significantly associated with patients’ mortality independent of oncological features such as tumor stage, disease status, or current provision of active anticancer therapy (Table 4).

## 3. Discussion

The management of cancer patients in the COVID-19 pandemic was initially shaped by presumptive evidence and precautionary measures [17]. Early reports of high mortality rates in patients with co-morbidities dictated a strict risk-stratification in the provision of cancer care. The acute pressure of rapidly escalating SARS-CoV-2 infection rates on healthcare systems adversely impacted on the ability to manage cancer patients in Western countries and, in most instances, reduced the capacity to escalate treatment beyond ward-based care. Whether cancer or active anticancer therapy per se influence severity and outcome from COVID-19 disease is the subject of concern and ongoing debate [18,19,20].

This study portrays a detailed account of the early SARS-CoV-2 outbreak in cancer patients in Europe, by surveying amongst the three hardest hit geographical areas in Europe: London (UK), North-West Italy, and Catalunya (Spain) during the initial phase of the outbreak; February and March 2020. Currently, there is a lack of published multi-center data sharing the initial impact on cancer patients during the early period of the COVID-19 outbreak in Europe. This report complements the published account from other geographical areas and will help to shape the ongoing optimal management of patients with cancer during the COVID-19 pandemic in Europe and globally, especially during the later stages of this outbreak.

The reported mortality rate of 28.9% in this study is similar to an initial retrospective case series from Wuhan which reported a mortality rate of 28.6% in cancer patients, with a worse disease course in those on active anticancer therapy [21]. This early report helped to shape the narrative that cancer and anticancer treatment was detrimental when combined with COVID-19 disease, however, the small numbers of patients with cancer in this report make the findings difficult to generalize to the larger population of patients with cancer. Although the high prevalence of lung cancer in the Wuhan cohort raises concerns over chronic airways disease as a confounding factor in estimating severity and mortality from COVID-19 in cancer, a study looking at mortality from COVID-19 in cancer patients during the initial outbreak in New York, USA, reported a similar mortality rate of 28% [7]. Taken together, this suggests that the mortality rate in cancer patients who contract COVID-19 was around 30%, at least during the initial period of the outbreaks in each country. Larger studies have been published subsequently including the CCC19 cohort study, which is a primarily North American study, which looked at more than 1000 cancer patients with COVID-19 and found that at time of analysis, 13% of patients had died [19]. This study involved many centers across North America with different trajectories of COVID-19 spread during the report period. This may explain the difference seen in mortality rates in this study compared to reports from the early phase of the outbreaks.

In the OnCovid study, presenting symptoms of COVID-19 in cancer patients were largely similar to those described in the general population, with high prevalence of fevers, cough, and dyspnea being often accompanied by a constellation of milder and less specific constitutional symptoms. The interval between onset of symptoms and presentation in our study (3.8 days) was shorter than the previously published mean of 7 days [22]: it is conceivable that that the pre-existing cancer-related symptomatic burden might have led to earlier presentation to acute services.

Interestingly, a quarter of patients appeared to have acquired COVID-19 whilst admitted to hospital for unrelated reasons, reinforcing the need for and robust infection control policies to limit nosocomial spread to the most vulnerable [21].

Approximately three quarters of patients in this study presented with or developed major complications from COVID-19, including high rates of respiratory failure requiring supplemental oxygen therapy (62.7%). Rate of progression to ARDS (24%) was within the limits of previously reported studies (19.6–41.8%) where <10% of patients carried a history of malignancy [23,24]. The significant proportion of complications and the relatively high co-morbid burden described in our patients contribute to explain our mortality estimate of 28.9%, which is similar to that reported for Wuhan cancer patients (28.6%) and others [4,7,20,21] but significantly higher compared to cancer-unselected hospitalized patients (4.3–11%) [1,23]. In keeping with previous studies in Covid-19 infected patients with and without cancer, smoking did not emerge as a clear independent predictor of mortality, underscoring the importance of age and comorbidities as more robust predictors of outcome [19,25,26].

Outcomes from SARS-Cov-2 are influenced by ceilings of medical care. However, mortality rates can be as high as 50% [24] even in selected patients treated within intensive care units [22]. In cancer patients, escalation beyond ward-based care is not always appropriate and requires careful case-by-case evaluation [27]. The decision to provide organ support to acutely ill cancer patients is made even harder in the context of a global pandemic, where saturation of clinical services imposes an often difficult prioritization of critical care resources in favor of younger and less co-morbid critically ill patients [28]. In our study, a minority of patients were admitted to intensive/sub-intensive care units and an even smaller proportion were intubated and ventilated. This precludes us from drawing definitive conclusions as to prognostic outlook and outcomes in this subpopulation: a point that will be investigated in future studies. An initial report covering the period from 26 March to 12 April 2020 from the TERAVOLT study which focuses on patients with thoracic malignancies and COVID-19 similarly reported a low admission rate to intensive care units [4].

In this study, no association was found between anticancer therapy and mortality. This may have been due to disease and therapeutic heterogeneity of our patients. Our findings are in keeping with a recent UK-based study, where 281 cancer patients receiving cytotoxic chemotherapy in a cohort of 800 cancer patients with a diagnosis of symptomatic COVID-19 did not appear to have different outcomes [20]. During the initial stages of the COVID-19 outbreak in Europe, due to the high risks of infection and limited medical resources, prioritization of certain anticancer treatments over others was routinely practiced [15]. Anticancer treatments with curative intent, higher chance of success, or leading to durable disease control were prioritized. By selecting the patients who were the fittest or most likely to benefit from treatment to continue anticancer therapy, we may have reduced the risks of more vulnerable or later stage cancer patients from becoming seriously ill or dying from COVID-19. Most patients in this study received systemic therapy with palliative intent: further analyses with greater sample size might help us clarify if the lack of association with mortality is preserved across chemotherapy regimens of various intensity. This will need further evaluation and comparison as the routine provision of anticancer treatment increases during the later stages of the COVID-19 pandemic.

This study acknowledges several limitations. As OnCovid is an observational study that relies on the collection of clinico-pathologic and outcome data generated during routine care, diagnostic pathways and therapeutic decisions were not standardized a priori across centers. As a result, some cases have incomplete documentation or missing laboratory/radiologic data. Thirdly, we elected to document outcomes of SARS-Cov-2 confirmed cases only. Due to the scarcity of RT-PCR testing at the beginning of the outbreak, available only to hospitalized patients in most centers, we might have failed to capture the whole spectrum of COVID-19 by artificially excluding asymptomatic patients or those with milder symptoms not requiring hospital assessment. Lastly, this interim analysis focuses on a 54.9% event rate including deaths and hospital discharges: whilst potentially leading to underestimation of clinical outcomes of interest, this is a shared limitation of most COVID-19 outcome studies so far, which have focused on early mortality rather than long-term outcomes. As such, our findings warrant verification in adequately powered long-term follow up studies. Despite the acknowledged limitations, our study is the largest and most geographically diverse in this disease area during the initial European outbreak of COVID-19: factors that broaden the generalizability of our results to the wider population of oncological patients requiring hospital assessment for COVID-19 and which will be compared with outcomes in cancer patients during the later phase of the pandemic.

## 4. Materials and Methods

### 4.1. Study Population, Setting, and Data Collection

Eligible patients had to be ≥18 years of age, have a confirmed diagnosis of SARS-CoV-2 infection by reverse-transcriptase polymerase chain reaction (RT-PCR) of a nasopharyngeal swab, and history of solid or hematologic malignancy, either active or in remission at the time of COVID-19 diagnosis. We generated a multi-center dataset of 204 patients consecutively diagnosed with COVID-19 in eight academic centers between 26 February and 1 April 2020. These included three sites in the United Kingdom (Imperial College; University College and Chelsea and Westminster Hospital, London), two in Italy (University of Piemonte Orientale, Maggiore della Carità Hospital, Novara; Policlinico San Matteo, Pavia), and three in Spain (Hospital Clinic; Barcelona, VHIO–Vall d’Hebron Institute of Oncology, Barcelona; Catalan Institute of Oncology, Girona). This study was granted central approval by the Health Research Authority in United Kingdom (20/HRA/1608) and by the corresponding ethics review board at each participating institution outside the UK. Prospective informed consent was waived by competent authorities due to the retrospective nature of the study and use of anonymized data. Clinical data including patients’ demographics, laboratory and radiologic results were collated from electronic medical records into a case report form designed using the Research Electronic Data Capture software (REDCap, Vanderbilt University, Nashville, TN, USA). To maintain confidentiality standards, each patient enrolled into the study was assigned a unique pseudonymization code through assignment of an identification number. We collected features of COVID-19 disease including presenting symptomatology, severity, requirement for and length of hospitalization, emergence of secondary complications. Outcomes from COVID-19 including recovery and mortality rates were documented. Multi-site access and data curation was coordinated by the Medical Statistics Unit in Novara, Italy. Data cut-off was 6 April 2020.

### 4.2. Study Definitions

The diagnosis of COVID-19 and description of the clinical syndromes associated with it including Acute Respiratory Distress Syndrome (ARDS) followed criteria published by the World Health Organization [29]. All patients recruited to this study were confirmed positive to SARS-CoV-2 following Real-time RT-PCR testing of nasopharyngeal swab samples using validated methodology [30]. Nosocomial SARS-CoV-2 transmission was defined in patients who developed symptoms and tested positive for COVID-19 whilst admitted for other reasons [23].

### 4.3. Statistical Analysis

Normally distributed data were presented as mean and standard deviation (SD), whereas data following a non-normal distribution were presented as median and interquartile range (IQR). Categorical variables were summarized as counts and percentages. Differences in medians were evaluated using Mann–Whitney’s U test and Wilcoxon Rank signed-rank test for pairwise comparisons. Associations between categorical variables were tested using Pearson’s Chi-square test or Fisher’s exact test as appropriate. Multivariable Cox proportional hazards models were used to adjust for differences in baseline factors, including age, comorbidities, disease stage, tumor status, and anticancer treatment. The hazard ratios (HRs) and the corresponding 95% CIs from the Cox proportional hazard model were calculated. A two-sided *p*-value < 0.05 was considered statistically significant. A random effect (frailty) term, incorporated as a shared frailty (frailty modeled between groups), accounts for the heterogeneity between centers. Analyses were performed using STATA software, version 14 (StataCorp. 2015. Statistical Software: Release 14.0. College Station, TX, USA) and SPSS version 25 (IBM Inc., Armonk, NY, USA).

## 5. Conclusions

In our study, we provide early but clinically important evidence that during the initial phase of COVID-19 in Europe, age and co-morbid burden dominated over oncological features in determining outcome from COVID-19. Neither tumor stage, disease activity, nor provision of active-anticancer therapy at COVID-19 diagnosis were enriched in non-survivors, whereas a striking and highly significant rise in mortality was seen for patients aged ≥65 and those with multiple co-morbidities.

Our findings will be validated in larger cohorts and with longer follow-up. The ongoing OnCovid study will allow for this and for comparisons between the early and later stages of the COVID-19 pandemic in terms of cancer patient outcomes after contracting COVID-19. Our initial study results lend further credence to the importance of comprehensive clinical risk-stratification during the COVID-19 pandemic to maintain cancer care a personalized effort. In this context, younger patients without co-morbidities should be carefully assessed for appropriateness of anticancer therapy and prioritized according to expected therapeutic benefit, whereas elderly and multiply co-morbid patients, in whom survival benefit from cancer therapy are outweighed by COVID-19 related morbidity and mortality, might be best managed with deferral or de-escalation of cancer therapy [17]. As SARS-Cov-2 infection continues to spread or recur, our data supports optimal, evidence-based risk-stratification centered on patients’ age and co-morbidities to protect the most vulnerable patients and avoid indiscriminate deferral and de-escalation of anticancer therapy to the detriment of oncological outcomes.

## Figures and Tables

**Figure 1 cancers-12-01841-f001:**
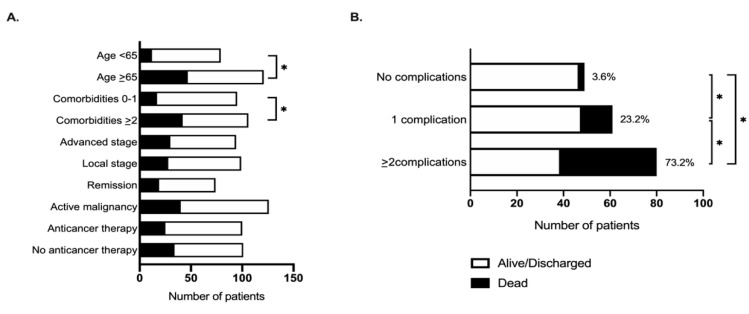
The relationship between mortality from COVID-19 and (**A**) baseline patient features and (**B**) complications from SARS-Cov-2 infection. * indicates *p* < 0.05.

**Table 1 cancers-12-01841-t001:** Patient characteristics.

Characteristic	Italy (*n* = 56)	Spain (*n* = 51)	UK (*n* = 97)	Total (*n* = 204)
Age (Years), mean ± SD	66.3 ± 14.0	69.4 ± 10.2	71.0 ± 13.3	69.3 ± 13.0
Age ≥ 65 years No. (%)	34 (60.7)	32 (62.7)	63 (65.0)	129 (63.2)
Sex No. (%)				
Male	32 (57.1)	31 (60.8)	64 (66.0)	127 (62.3)
Female	24 (42.9)	20 (39.2)	33 (34.0)	77 (37.7)
Smoking history No. (%)				
Never smoker	24 (42.9)	21 (41.2)	34 (35.0)	79 (38.7)
Current/former smoker	25 (44.6)	28 (54.9)	59 (60.8)	112 (54.9)
Unknown	7 (12.5)	2 (3.9)	4 (4.1)	13 (6.4)
Cancer type No. (%)				
Genito-urinary	7 (12.5)	11 (21.6)	25 (25.8)	43 (21.1)
Lung	6 (10.7)	12 (23.5)	18 (18.6)	36 (17.7)
Gastrointestinal	8 (14.3)	6 (11.8)	14 (14.4)	28 (13.7)
Breast	8 (14.3)	8 (15.7)	11(11.3)	27 (13.2)
Gynecological	8 (14.3)	3 (5.9)	7 (7.2)	13 (6.4)
Gastro-esophageal	3 (5.4)	2 (3.9)	5 (5.1)	10 (4.9)
Hepatobiliary	2 (3.6)	1 (2.0)	6 (6.2)	9 (4.4)
Head and neck	1 (1.8)	2 (3.9)	4 (4.1)	7 (3.4)
Skin	-	2 (3.9)	1 (1.0)	3 (1.5)
Other	2 (3.6)	3 (5.9)	1 (1.0)	6 (2.9)
Hematological malignancies	16 (28.6)	-	3 (3.1)	19 (9.8)
Tumor Stage * No. (%)				
Localized	16 (43.2)	15 (29.4)	63 (64.9)	94 (50.8)
Metastatic	20 (54.0)	30 (58.8)	32 (33.0)	82 (44.3)
Unknown	1 (2.7)	6 (11.8)	2 (2.1)	9 (4.9)
Prior treatments No. (%)				
Surgery	19 (33.9)	28 (54.9)	49 (50.5)	96 (47.1)
Adjuvant/neoadjuvant chemotherapy	7 (12.5)	17 (17.5)	25 (25.8)	49 (24.0)
Palliative systemic therapy No. (%)	21 (37.5)	18 (35.3)	22 (22.7)	61 (29.9)
Chemotherapy	18 (85.7)	7 (38.9)	13 (59.1)	38 (62.3)
Immunotherapy	4 (19.0)	-	1 (4.5)	5 (8.2)
Endocrine therapy	2 (9.5)	6 (33.3)	5 (22.7)	13 (21.3)
Target therapy	2 (9.5)	7 (38.9)	4 (18.2)	13 (21.3)
Curative systemic therapy ** No. (%)	11 (68.7)	-	2 (66.7)	13 (68.4)
Radiotherapy No. (%)	18 (32.1)	22 (43.1)	25 (25.8)	65 (31.9)
Prior lines of palliative therapy * No. (%)				
1	13 (65.0)	8 (50.0)	9 (50.0)	30 (55.6)
2	3 (15.0)	1 (6.3)	3 (16.7)	7 (13.0)
≥3	4 (20.0)	7 (43.7)	6 (33.3)	17 (31.4)
Co-morbidities No. (%)				
Hypertension	20 (35.7)	23 (45.1)	45 (46.4)	88 (43.1)
Diabetes	10 (17.9)	10 (19.6)	26 (26.8)	46 (22.6)
Cardiovascular disease	8 (14.3)	11 (22.5)	25 (25.8)	44 (21.5)
Chronic pulmonary disease	7 (12.5)	12 (23.5)	15 (15.5)	34 (16.7)
Chronic kidney disease	2 (3.6)	12 (23.5)	18 (18.6)	32 (15.7)
Cerebrovascular disease	1 (1.8)	2 (3.9)	13 (13.4)	16 (7.8)
Dementia	-	3 (5.9)	9 (9.3)	12 (5.9)
Peripheral vascular disease	1 (1.8)	3 (5.9)	4 (4.1)	8 (3.9)
Liver impairment	-	3 (5.9)	3 (3.1)	6 (2.9)
Immunosuppression	-	2 (3.9)	3 (3.1)	5 (2.5)
Other	6 (10.7)	11 (21.6)	32 (33.0)	49 (24.0)
Number of comorbidities No. (%)				
0	17 (30.4)	9 (17.7)	17 (17.5)	43 (21.1)
1	16 (28.6)	18 (35.3)	20 (20.6)	54 (26.5)
2	16 (28.6)	7 (13.7)	24 (24.7)	47 (23.0)
≥3	7 (12.5)	17 (33.3)	36 (37.1)	60 (29.4)
Ongoing anticancer therapy at COVID-19 Diagnosis No. (%)	35 (62.5)	31 (60.8)	36 (37.1)	102 (50.0)
COVID-19 symptoms No. (%)				
Fever	40 (71.4)	41 (80.4)	55 (56.7)	136 (66.7)
Cough	22 (39.3)	34 (66.7)	63 (65.0)	119 (58.3)
Dyspnea	20 (35.7)	14 (27.5)	47 (48.5)	81 (39.7)
Fatigue	9 (16.1)	24 (47.1)	18 (18.6)	51 (25.0)
Myalgia	4 (7.1)	9 (17.7)	11 (11.3)	24 (11.8)
Diarrhea	4 (7.1)	8 (15.7)	8 (8.3)	20 (9.8)
Coryzal symptoms	-	6 (12.0)	5 (5.2)	11 (5.4)
Nausea or vomiting	3 (5.4)	2 (3.9)	5 (5.2)	10 (4.9)
Sore throat	-	2 (3.9)	4 (4.1)	6 (3.0)
Headache	-	3 (5.9)	2 (2.1)	5 (2.5)
Dysgeusia	1 (1.8)	1 (2.0)	1 (1.0)	3 (1.5)
Anosmia	2 (3.6)	-	1 (1.0)	3 (1.5)
Other (confusion, delirium)	9 (16.1)	5 (9.8)	23 (23.7)	37 (18.1)
Hospitalization rate No. (%)	44 (78.6)	50 (98.0)	92 (94.8)	186 (91.2)
Admission to ICU No. (%)	13/44 (30.0)	2/50 (4.0)	21/92 (22.8)	36/186 (19.4)
COVID-19 Specific Treatments No. (%)				
Antibiotics	33 (58.9)	47 (92.2)	58 (59.8)	138 (67.7)
Hydroxychloroquine	20 (60.6)	38 (80.9)	49 (84.5)	107 (77.5)
Lopinavir/ritonavir	27 (81.8)	43 (91.5)	14 (24.1)	84 (60.9)
Corticosteroids	3 (9.1)	31 (66.0)	7 (12.1)	41 (29.7)
Remdesivir	3 (9.1)	5 (10.6)	7 (12.1)	15 (10.9)
Tocilizumab	8 (24.2)	1 (2.1)	-	9 (6.5)
Others	-	5 (10.6)	-	5 (3.6)
Oxygen therapy No. (%)	3 (9.1)	22 (46.8)	3 (5.2)	28 (20.3)
Mechanical ventilation No. (%)	32 (57.1) 3 (5.4)	31 (60.8) -	65 (67.0) 15 (15.5)	128 (62.8) 18 (8.8)
COVID-19 complications No. (%)	32 (57.1)	31 (60.7)	78 (81.2)	141 (69.1)
Acute respiratory failure	32 (57.1)	31 (60.7)	65 (67.0)	128 (62.7)
ARDS ^§^	11 (19.6)	17 (33.3)	21 (20.7)	49 (24.0)
Acute kidney injury	1 (1.8)	10 (19.6)	14 (14.4)	25 (12.3)
Secondary infection	4 (7.1)	2 (3.9)	14 (14.4)	20 (9.8)
Acute cardiac injury	-	-	3 (3.1)	3 (1.5)
Acute liver injury	-	1 (2.0)	2 (2.1)	3 (1.5)
Others	-	3 (5.9)	12 (12.4)	15 (7.4)

* solid tumors; ** haematological malignancies; ^§^ ARDS Acute Respiratory Distress Syndrome.

**Table 2 cancers-12-01841-t002:** Routine laboratory investigations at COVID-19 diagnosis and at last oncological follow-up.

Laboratory Findings	COVID-19 Diagnosis ^1^ Median (IQR) *n* = 193	Pre COVID-19 ^2^ Median (IQR) *n* = 128	*p* Value
Albumin	32.0	40.0	*p* < 0.0001
(g/L)	(28.0–37.0)	(37.0–43.0)	
Bilirubin	8.8	9.7	*p* = 0.13
(µmol/L)	(5.2–13.8)	(6.6–13.8)	
Alanine aminotransferase	30.5	21.0	*p* < 0.0001
(IU/L)	(20.0–47.0)	(16.0–33.0)	
Sodium	136.0	139.0	*p* < 0.0001
(mEq/L)	(134.0–139.0)	(137.0–141.0)	
Urea	7.2	6.8	*p* = 0.29
(mmol/L)	(4.8–10.7)	(4.8–9.6)	
Creatinine	80.5	74.3	*p* = 0.13
(µmol/L)	(56.6–109.7)	(61.1–92.0)	
Hemoglobin	115.0	118.0	*p* = 0.002
(g/L)	(101.0–132.0)	(102.0–133.0)	
Leukocyte count	6.8	6.7	*p* = 0.34
(cells 10^3^/mm^3^)	(4.3–10.0)	(4.9–8.9)	
Neutrophil count	5.4	4.5	*p* = 0.06
(cells 10^3^/mm^3^)	(3.0–8.0)	(2.8–6.4)	
Lymphocyte count	0.7	1.4	*p* < 0.0001
(cells 10^3^/mm^3^)	(0.5–1.1)	(1.0–1.9)	
Platelet count	188.0	234.0	*p* < 0.0001
(cells 10^6^/mm^3^)	(141.0–273.0)	(182.0–295.0)	
C-reactive protein	91.0	8.1	*p* < 0.0001
(mg/L)	(36.9–181.4)	(1.6–25.9)	

^1^ Obtained at the time of hospital assessment for COVID-19. For each variable data were available for (*n*=): Albumin (*n* = 105), Bilirubin (*n* = 134), ALT (*n* = 178), Na (*n* = 186), Urea (*n* = 80), Creatinine (*n* = 187), Hb (*n* = 191), Leukocytes (*n* = 193), Neutrophils (*n* = 192), Lymphocytes (*n* = 192), Platelets (*n* = 193), CRP (*n* = 169). ^2^ Obtained at last cancer follow up visit or anticancer treatment cycle. For each variable data were available for (*n*=): Albumin (*n* = 63), Bilirubin (*n* = 103), ALT (*n* = 115), Na (*n* = 109), Urea (*n* = 49), Creatinine (*n* = 129), Hb (*n* = 127), Leukocytes (*n* = 126), Neutrophils (*n* = 128), Lymphocytes (*n* = 127), Platelets (*n* = 127), CRP (*n* = 38), Ferritin (*n* = 41).

**Table 3 cancers-12-01841-t003:** The relationship between patient characteristics and mortality from COVID-19: univariable analysis.

Characteristic	Alive/Discharged	Dead	HR (95% CI)
Age, years No. (%)			
<65	62 (42.8)	12 (20.3)	1.0
≥65	83 (57.2)	47 (79.7)	2.6 (1.4–5.0)
Comorbidities No. (%)			
0–1	80 (55.2)	17 (28.8)	1.0
≥2	65 (44.8)	42 (71.2)	2.4 (1.3–4.2)
Smoking history No. (%)			1.0 1.3 (0.8–2.3)
Lifetime non-smoker	59 (40.7)	20 (33.9)	
Ex/active smoker	75 (51.7)	37 (62.7)	
Unknown	11 (7.6)	2 (3.4)	
Tumor stage No. (%)			1.0 1.4 (0.8–2.6)
Local/loco-regional	67 (52.3)	27 (47.4)	
Advanced	53 (41.4)	29 (50.9)	
Unknown	8 (6.2)	1 (1.8)	
Tumor status No. (%)			
Remission/no measurable disease	57 (39.3)	19 (32.2)	1.0
Active malignancy	88 (60.7)	40 (67.8)	1.4 (0.8–2.5)
Anticancer therapy No. (%)			
Yes	77 (53.1)	25 (42.4)	1.0
No	68 (46.9)	34 (57.6)	1.1 (0.6–1.9)
Cytotoxic chemotherapy No. (%)			
Yes	46 (31.7)	16 (27.1)	1.0
No	99 (68.3)	43 (72.9)	1.0 (0.6–1.9)
COVID-19 therapy No. (%)			
Yes	103 (71.0)	35 (59.3)	1.0
No	42 (29.0)	24 (40.7)	1.6 (0.9–2.8)
Temperature at COVID-19 diagnosis (median, IQR)	37.9 (36.9–38.3)	38.0 (36.9–38.6)	*p* = 0.8
Duration of symptoms prior to COVID-19 diagnosis (median, IQR)	2.5 (1.0–6.0)	2.5 (0–4.0)	*p* = 0.8
Interval from last anticancer treatment to COVID-19 diagnosis (median, IQR)	12.0 (7.0–22.5)	16.0 (8.0–34.0)	*p* = 0.7
Laboratory findings at COVID-19 diagnosis (median, IQR)			
Albumin (g/L)	3 (27–36)	34 (29–38)	*p* = 0.4
Bilirubin (µmol/L)	8.6 (5.2–13.0)	9 (6–15.5)	*p* = 0.6
Alanine aminotransferase (IU/L)	28 (18–42)	32 (23–51)	*p* = 0.06
Sodium (mEq/L)	136 (134–139)	136 (133–138)	*p* = 0.4
Urea (mmol/L)	6.7 (4.3–9.8)	10 (6.3–16)	*p* = 0.4
Creatinine (µmol/L)	72.6 (54.9–98.2)	95.1 (72–141)	*p* < 0.001
Hemoglobin (g/L)	116 (101.5–132)	109 (99–129)	*p* = 0.2
Leukocyte count (cells 10^3^/mm^3^)	6.1 (4.0–9.5)	8.2 (5.3–11.3)	*p* = 0.7
Neutrophil count (cells 10^3^/mm^3^)	4.5 (2.8–7.4)	6.5 (3.6–10.0)	*p* = 0.7
Lymphocyte count (cells 10^3^/mm^3^)	0.8 (0.5–1.1)	0.6 (0.5–1.0)	*p* = 0.3
Platelet count (cells 10^6^/mm^3^)	190 (145–273)	173 (139–278)	*p* = 0.9
C-reactive protein (mg/L)	69 (22.9–154.3)	143.3 (70–260.6)	*p* = 0.002
Lactate dehydrogenase (IU/L)	363 (251–524)	447 (276–596)	*p* = 0.04
D-dimer (ng/mL)	655.5 (463.5–1514)	2100 (761–3387)	*p* = 0.2
Ferritin (ng/mL)	650 (215–1297)	421 (289–1010)	*p* = 0.7

**Table 4 cancers-12-01841-t004:** The relationship between patient characteristics and mortality from COVID-19: multivariable analysis.

Characteristic	Alive/Discharged	Dead	HR (95%CI)
Age No. (%)			
<65	62 (42.8)	12 (20.3)	1.0
≥65	83 (57.2)	47 (79.7)	2.2 (1.0–4.6)
Comorbidities No. (%)			
0–1	80 (55.2)	17 (28.8)	1.0
≥2	65 (44.8)	42 (71.2)	1.9 (1.0–3.6)
Tumor stage No. (%)			1.0 1.5 (0.7–3.2)
Local/loco-regional	67 (52.3)	27 (47.4)	
Advanced	53 (41.4)	29 (50.9)	
Unknown	8 (6.2)	1 (1.8)	
Tumor status No. (%)			
Remission/no measurable disease	57 (39.3)	19 (32.2)	1.0
Active malignancy	88 (60.7)	40 (67.8)	1.3 (0.6–2.8)
Anticancer therapy No. (%)			
Yes	77 (53.1)	25 (42.4)	1.0
No	68 (46.9)	34 (57.6)	1.3 (0.7–2.6)

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
