# Peer review of "Presenting Features and Early Mortality from SARS-CoV-2 Infection in Cancer Patients during the Initial Stage of the COVID-19 Pandemic in Europe"

_cancers, 2020, doi:10.3390/cancers12071841_

Round 1

Reviewer 1 Report

The text is interesting from a clinical-methodological
point of view. The series of patients is well described from both
an oncological and infectious point of view.

1) Did cancer patients who survived build anti COVID-19
antibodies?

2) Among the tests performed, ferritin is not indicated,
why?

3) mortality among smokers and non-smokers was not
significant, can the authors comment on this aspect?

4) cytostatic therapy was not associated
with statistically significant mortality,
can the authors comment on this aspect? Intensity
of chemotherapy?

Author Response

We are thankful for the positive feedback on our work.

In answer to the reviewer's comments:

1) Did cancer patients who survived build anti COVID-19 antibodies?

We acknowledge the reviewer’s comment. Unfortunately antibody testing was not routinely performed during the time period covered – however may be addressed in a later paper as antibody testing becomes part of routine diagnostic procedures.

2) Among the tests performed, ferritin is not indicated, why?

Ferritin values were available on a minority of patients at Covid-19 diagnosis (n=41). We have added Ferritin levels to the analysis of predictors of mortality in Table 3.

3) mortality among smokers and non-smokers was not significant, can the authors comment on this aspect?

We have underscored the importance of this finding in the discussion, which resonates with evolving evidence suggesting lack of influence of smoking in the prognosis of Covid-19 – both in patients with and without cancer at the moment of SARS-Cov-2 infection. We have updated the references to this effect.

4) cytostatic therapy was not associated with statistically significant mortality,

can the authors comment on this aspect? Intensity of chemotherapy? 

We are thankful for the suggestion. A number of studies have now shown that systemic chemotherapy does not influence the outcome of SARS-Cov-2 infection in terms of severity and risk of mortality. We have underscored the importance of this finding in the discussion and provided references in support. In our paper, the majority of patients were undergoing treatment with palliative intent and it is therefore difficult, due to the relatively small sample size and heterogeneity in oncological diagnosis, to trace a parallel between intensity of chemotherapy and deteriorating prognosis in the context of Covid-19 infection. As the OnCovid registry initiative continues to grow, we will evaluate this aspect in going forward.

Reviewer 2 Report

An important contribution in the field, given the following:

  • multicenter feature of the study,
  • the included population (cancer patients as vulnerable population),
  • significant results with risk stratification by age and comorbidities,
  • the finding that anticancer therapy did not influence prognostic and therefore it is recommended not to stop anticancer therapy in these patients,
  • pertinent discussions and
  • emphasis of the weak and strong points of the study.

Minor suggestion

Please avoid such frequent use of personal expression (we, our); please use / replace with impersonal expression (they, it, etc.) that sounds more professional (i.e. not "our patients" but "the or these patients"; L319 "In our study, we provide..." replaced with "In this study/research/paper/article (as the authors chose) it was provided...."); page 9 - 9 times "our" usage; L227 "our mortality"? it does not sounds good; etc.). Please revise the entire manuscript in this regard.

Author Response

We are thankful for the positive feedback from reviewer 2. We have followed their advice and modified the use of personal expressions in the text and changed the style of the manuscript to an impersonal tone.